# Identification of Ovarian High-Grade Serous Carcinoma with Mitochondrial Gene Variation

**DOI:** 10.3390/ijms26031347

**Published:** 2025-02-05

**Authors:** Jesus Gonzalez Bosquet, Vincent Wagner, Andrew Polio, Katharine E. Linder, David P. Bender, Michael J. Goodheart, Brandon M. Schickling

**Affiliations:** 1Department of Obstetrics and Gynecology, University of Iowa, Iowa City, IA 52242, USAandrew-polio@uiowa.edu (A.P.); katharine-linder@uiowa.edu (K.E.L.); david-bender@uiowa.edu (D.P.B.); brandon-schickling@uiowa.edu (B.M.S.); 2Holden Comprehensive Cancer Center, University of Iowa Hospitals and Clinics, Iowa City, IA 52242, USA

**Keywords:** genetic variation, ovarian cancer, prediction model, whole-exome sequencing—WES, RNA sequencing—RNAseq

## Abstract

Women diagnosed with advanced-stage ovarian cancer have a much worse survival rate than women diagnosed with early-stage ovarian cancer, but the early detection of this disease remains a clinical challenge. Some recent reports indicate that genetic variations could be useful for the early detection of several malignancies. In this pilot observational retrospective study, we aimed to assess whether mitochondrial DNA (mtDNA) variations could discriminate the most frequent type of ovarian cancer, high-grade serous carcinoma (HGSC), from normal tissue. We identified mtDNA variations from 20 whole-exome sequenced (WES) HGSC samples and 14 controls (normal tubes) using the best practices of genome sequencing. We built prediction models of cancer with these variants, with good performance measured by the area under the curve (AUC) of 0.88 (CI: 0.74–1.00). The variants included in the best model were correlated with gene expression to assess the potentially affected processes. These analyses were validated with the Cancer Genome Atlas (TCGA) dataset, (including over 420 samples), with a fair performance in AUC terms (0.63–0.71). In summary, we identified a set of mtDNA variations that can discriminate HGSC with good performance. Specifically, variations in the *MT-CYB* gene increased the risk for HGSC by over 30%, and *MT-CYB* expression was significantly decreased in HGSC patients. Robust models of ovarian cancer detection with mtDNA variations could be applied to liquid biopsy technology, like those which have been applied to other cancers, with a special focus on the early detection of this lethal disease.

## 1. Introduction

The early detection of ovarian cancer remains elusive. Although patients with early stages have a 5-year survival rate exceeding 90%, over 70% of patients are diagnosed with advanced disease with 5-year overall survival of around 40% [1,2]. The data are compelling: early diagnosis of ovarian cancer confers a better prognosis and survival. Unfortunately, no current method of screening has proven effective at detecting ovarian cancer at an earlier stage [3].

Recently, there have been some reports indicating the possibility of using genomic variations in the detection of several malignancies, including variations in mitochondrial DNA (mtDNA) [4,5,6,7]. While mtDNA genetic variation may not cause direct transcriptional or translational consequences, it is possible to use these variations to create models that would discriminate cancer cells from benign cells. Cell-free DNA (cfDNA) is DNA not contained within cells, and present in other bodily fluids. cfDNA preserves the genetic make-up of the tissue of origin and permits the genetic characterization of tumors by non-invasive procedures, referred to as *liquid biopsy* [8]. There have been recent reports on the use of cfDNA variations/mutations to detect cancer, even in the early stages [9,10]. For example, in lung cancer, liquid biopsy is used for prognosis and treatment response using mutations in cfDNA [11,12], and it has been used to create a new test for early diagnosis [10,12]. Likewise, liquid biopsy has been used successfully in the diagnosis, prognosis, and treatment management of *KRAS*-mutated colon cancer [13]. In breast cancer, significant efforts are underway to determine the use of liquid biopsy in one of the most aggressive forms of this disease: triple-negative cancer [14]. Therefore, it seems likely that mitochondrial variations found in cfDNA could also be used for the (early) diagnosis of ovarian cancer. To assess this possibility, we leveraged a well-annotated biobank of ovarian cancer and normal fallopian specimens, and determined and compared genomic variations in these samples.

The aim of this pilot study was to assess whether mtDNA variations could discriminate the most frequent type of ovarian cancer, high-grade serous carcinoma (HGSC), from normal tissue. We validated the prediction models of HGSC with mtDNA variations in different platforms and an independent dataset: the Cancer Genome Atlas (TCGA) HGSC dataset. Additionally, we correlated significant mtDNA variants with HGSC gene expression and determined if the associated gene pattern was related to the clinical outcomes of HGSC.

## 2. Results

### 2.1. mtDNA Single Nucleotide Variation (SNV)

Whole-exome sequencing (WES) of DNA extracted from all cases (HGSC, N = 20) and the controls (fallopian tube, N = 14) detected 393 variants in 37 mitochondrial genes. All variants, including their locations and descriptions, can be found in Appendix A. Initially, in the discovery phase of the study, we introduced all these mtDNA SNVs in a LASSO regression multivariable analysis. First, we assessed the penalty parameter (or lambda λ) to introduce the mtDNA SNVs in the model using bootstrapping to minimize overfitting. The resulting λ (0.136) was smaller than both the minimum and 1-standard error lambda usually suggested by the *glmnet* package (lambda.min = 0.176 and lambda.1se = 0.255, respectively). The lambda values are represented in Figure 1B by the dotted lines. Applying these settings to the model resulted in the selection of the five most informative variants for HGSC prediction (Figure 1A). Variants in the mitochondrial genes *MT-ATP8*, *MT-ND5*, and *MT-CYB* increased the risk for HGSC (Odds Ratio, OR > 1), while those in genes *MT-TP* and *MT-CO1* were protective (OR < 1). The performance of this prediction model, measured by the AUC, was 0.88 (95% CI: 0.74–1.00) (Figure 1B).

### 2.2. Correlation of Significant mtDNA Variants with Gene Expression

We compared gene expressions between HGSC samples (N = 112) and controls (N = 12). RNA sequencing (RNAseq) experiments resulted in 3382 transcripts differentially expressed (out of 61,851 total), with an adjusted *p*-value < 0.005 to account for multiple comparisons (Figure 2A). All significant genes are listed in Appendix A. Of all 37 mitochondrial genes, 13 had significant differential expressions between cases and controls: *MT-RNR1*, *MT-TS1*, *MT-TM*, *MT-ND3*, *MT-TP*, *MT-TK*, *MT-TS2*, *MT-TV*, *MT-CO2*, *MT-CO1*, *MT-TL1*, *MT-CYB*, and *MT-TY*. These 13 significant mitochondrial genes were introduced in a multivariable regression analysis resulting in four genes independently associated with ovarian cancer risk (Figure 2B).

*MT-CYB* mitochondrial gene variation was selected in the prediction model of ovarian cancer, conferring a higher risk for HGSC. Additionally, this gene’s expression was significantly reduced in women with HGSC (Figure 2B). To assess the repercussions of *MT-CYB* alterations in ovarian cancer on other genes and processes, we first determined which genes were significantly affected by these alterations, and then we estimated the effect in different biological processes using enrichment pathway analysis.

First, we correlated *MT-CYB* mitochondrial gene expression with all 3382 transcripts that were significant in the differential gene expression between HGSC and controls to discard genes not being significantly changed (or background noise). We identified 1728 genes (out of 3382) that were significantly correlated with variation in *MT-CYB* (FDR-adjusted *p*-value < 0.005). In Figure 2C, we present the top 20 genes. Then, we correlated *MT-CYB* gene expression (also significant in the multivariable analysis) with all significantly expressed genes (3382) and found 364 genes significantly correlated (FDR-adjusted *p*-value < 0.005). Figure 2D shows the top 15. There were 202 common genes correlated with variation and expression of the *MT-CYB* gene. In total, there were 1890 unique genes, the expression of which correlated with variation in the *MT-CYB* gene and/or with *MT-CYB* gene expression. Of note, the expression of 202 of these genes simultaneously correlated with variation and expression of the *MT-CYB* gene (Appendix A).

### 2.3. Pathway Enrichment Analysis of Correlated Genes

The pathway enrichment analysis using the KEGG database was performed with all unique 1890 genes, the expression of which was significantly correlated with *MT-CYB* genetic variation and/or expression. Significant pathways are listed in Table 1.

The top-ranked pathway is oxidative phosphorylation (OXPHOS), part of the energy metabolism of the cell. Enzymes of this pathway oxidize nutrients to release energy in the form of adenosine triphosphate (ATP). OXPHOS consists of five protein complexes I-V and takes place in the mitochondria (Figure 3).

### 2.4. Prediction Model Training, Validation and Testing

The five SNV in the mtDNA selected in the discovery phase by the multivariable regression model were used to train and validate the models with two different analytical platforms: (1) *LASSO* in the R (v4.4.1) environment (Figure 4a,b) and (2) the *MATLAB* (vR2023b) analytic platform (Figure 4c,d). The performance of the validated model using only the five most informative University of Iowa (UI) data was 0.91 (CI: 0.82–1.00) when we used *LASSO* (Figure 4a) and 0.95 (CI: 0.87–1.00) for the *MATLAB* platform (Figure 4e). The performance in the validation set was slightly better than in the training dataset (Figure 1B) because these analyses were performed with selected variables in the same dataset.

After downloading, processing, and extracting genetic variations from WES files from the HGSC TCGA dataset, we selected the same five variants included in the trained and validated prediction model of HGSC. Testing of the *LASSO* model with TCGA data and the *pROC* package (v1.18.5) resulted in a performance of 0.57 (CI: 0.53–0.62; Figure 4d). Testing the *MATLAB* model on TCGA data performed better, with an AUC of 0.71 and a 95% CI of 0.53–0.88 with partial overlap with the initial model and its validation analysis CIs (Figure 4f).

### 2.5. Correlation Analysis Validation

RNAseq experiments from the TCGA HGSC dataset were downloaded and processed. Out of the 3382 differentially expressed genes between HGSC and control samples, the TCGA dataset had information for 2716 transcripts for 423 samples. Out of the 1728 genes that were significantly correlated with variation in *MT-CYB* in the UI dataset, 1370 transcripts had information in TCGA set (were common for both datasets). There was a fair agreement, with an AUC of 0.68 (CI: 0.66–0.71, Figure 4g), between UI genes and TCGA genes correlated with *MT-CYB* variation, with 936 (68.3%) with correlations in the same direction (positive, r > 0, or negative, r < 0; Figure 4h). Additionally, out of the 364 genes significantly correlated between *MT-CYB* expression and the expression of all other genes in the UI set, 303 had information in the TCGA dataset (were common for both datasets). The agreement was lower, with an AUC of 0.63 (CI: 0.58–0.69, Figure 4i). The expression of 111 genes (36.6%) was correlated in the same direction in both databases (positive, r > 0, or negative, r < 0; Figure 4j).

## 3. Discussion

The primary goal of this pilot study was to identify genetic mitochondrial variations that could differentiate HGSC from normal tubal tissue. We used WES to identify mtDNA variations that were used to create prediction models. Then, these models were validated in an independent database and on different platforms with different analytics. Our model with five SNV from five different mitochondrial genes had a performance, measured by the AUC, of 0.88 (95% CI of 0.74–1.00). The validation of this model in different analytics platforms using the same UI dataset was consistent with the initial model and with AUCs of 0.91 and 0.95. Testing of this prediction model in an independent dataset, TCGA, and using varied machine learning analytics achieved performances of up to 0.71, with 95% CI (0.53–0.88) that overlaps, in part, with the initial prediction model of AUC CI. This is a fair-performing model [15] that, by itself, may not be sufficient to create a diagnostic tool, but in combination with other predictive models [16,17,18], may achieve the goal of predicting ovarian cancer accurately and robustly. The lower performance in the testing set from TCGA could be due, at least in part, to the use of control samples collected from normal tissues of the same patients from whom corresponding tumor samples were obtained and not from independent normal tubes (as in the UI dataset). How different control origins will impact model performance is unknown, mainly due to the peculiarities of the mtDNA to acquire mutations even in normal tissues (discussed later). Additionally, underlying differences in the genetic substructures of TCGA and UI populations may affect the model testing performance in TCGA [19]. Only variation analysis of large and diverse populations, with similar genetic admixture, would be completely comparable.

The ovarian cancer prediction model included five mtDNA SNVs. Specifically, variation in the *MT-CYB* gene increased the risk for HGSC by over 30%, and *MT-CYB* expression was significantly decreased in HGSC patients between cancer samples and controls. These differences were upheld in the multivariable analysis after adjusting for the expression of other mitochondrial genes. The *MT-CYB* variation was only seen in cancer specimens, which would make it a good potential prediction marker for ovarian cancer. In TCGA data, less than 10% of the controls had an *MT-CYB* variation, and in more than half of them, the variation was found in both the HGSC sample and the corresponding normal control. Cytochrome b (*MT-CYB*) is the only mtDNA encoded subunit of the respiratory Complex III (ubiquinol:ferrocytochrome c oxidoreductase complex). Complex III is located within the mitochondrial inner membrane, and is the second enzyme in the electron transport chain of mitochondrial oxidative phosphorylation. It catalyzes the transfer of electrons from ubiquinol (reduced Coenzyme Q10) to cytochrome c and utilizes the energy to translocate protons from inside the mitochondrial inner membrane to outside and is a highly evolutionarily conserved, hydrophobic protein containing two heme groups [20,21]. *MT-CYB* mutations have been implicated in ovarian carcinoma previously [22]. It seems that mutations in the *MT-CYB* gene are involved in remodeling mitochondrial metabolism with increases in the production of reactive oxygen species (ROS) within cancer cells [23]. This may be part of ovarian cancer cell adaptation to hypoxic conditions [24]. Consistent with this, is the observation of elevated mtDNA mutations from primary OC lesions to metastasis lesions, suggesting the presence of driver mutations conferring OC metastatic advantage [22]. The mitochondrial gene *MT-ATP8* harbored the most significant variant in our predictive model of cancer. Interestingly, recent reports have shown that fragments of this gene, *MT-ATP8*, are present in both blood plasma and exosomes in particularly aggressive types of lung cancer, making it a potential candidate for liquid biopsy diagnostics [25].

mtDNA is especially idiosyncratic because it is highly susceptible to acquiring mutations from ROS, even within normal tissues. Therefore, multiple subpopulations of mtDNA could arise during the mitochondria lifespan; this is the phenomenon known as heteroplasmy [26]. Heteroplasmy describes the situation in which two or more mtDNA variants exist within the same cell. These variants can occur either in the normal or in the tumoral tissues and can vary from cell to cell. Therefore, up to 72% of the reported tumor-specific mtDNA mutations are also found in non-tumor cells (germline variants) of healthy subjects [27]. Heteroplasmy is the reason why we decided to use all variants, without considering origin, to build our model of ovarian cancer prediction. The model itself discriminated which variants were informative for ovarian cancer risk, even if they are derived from mutations in normal tissue.

There are several classes of clinically relevant mtDNA variants that can contribute to neoplastic transformation [28]. Somatic and germline mtDNA mutations have been associated with several cancers, like renal, colon, head and neck, pancreatic, breast, ovarian, prostate, and bladder cancers [28,29]. In prostate cancer, the total burden of acquired mtDNA variants may be a biomarker for tumorigenicity [30]. A mtDNA cytochrome *c* oxidase subunit 1 (*MT-CO1*) variant has been associated with protection for ovarian cancer, as we also observed [31]. However, the functional consequences of such variations are less known. Using co-expression of mtDNA and nuclear DNA (nDNA) genes, researchers found OXPHOS pathways as the top-ranked enriched pathway in 8 of 13 cancer types examined [32]. In the present study, the OXPHOS pathway was also the most significant pathway in the enrichment analysis of the differentially expressed genes between HGSC samples and normal tubes. Furthermore, some of the top transcripts associated with mtDNA variation are associated with oxidizing neurotransmitters at the outer mitochondrial membrane (*MAOA*) that are responsible for behavioral changes [33], or encode tumor-associated proteins (like *CHIC1*) [34], or preserve the homeostasis of the cell, like *GPR89A* involved in intracellular pH regulation [35].

Comparing gene expressions of normal fallopian tubes and HGSC samples seems to be the most robust approach to identify these functional consequences in contraposition to comparisons to other normal tissue types, even from the genital tract [36]. Unlike eukaryotic nDNA, mitochondria can contain many copies of mtDNA. Mitochondrial replication, transcription, and translation are coordinated by nDNA-encoded proteins and mitochondrial-encoded rRNAs, tRNAs, and adapt to external stimuli and stressors [37]. Cancer cells may deregulate these mechanisms for survival benefit. Understandings of mechanisms could potentially be used as leverage to develop targeted therapies [38]. Some studies have raised the possibility that the tumor microenvironment may play an important role in regulating the mitochondrial gene expression of tumor cells [38]. For example, in our study, there were top transcripts associated with mtDNA variations that are known long non-coding RNAs (lncRNAs) and micro-RNAs described to be associated with epigenetic regulation in a variety of cancers, like *MIR4423*, *AC010280.1*, *AC010280.3*, *AL121839.2*, and *AC078883.2* [39,40,41,42,43]. However, there are still knowledge gaps in how changes within these processes may affect specific cancers in specific contexts. Furthermore, the transfer of mtDNA sequences into the nDNA, or *numtogenesis*, seems to be activated in certain tumors, like colorectal cancer, and may affect prognosis [44]. Though the mechanisms are not clear, it seems that mtDNA could be inserted within the nDNA of tumor suppressor genes, disrupt cellular pathways, and promote tumorigenesis [44]. Thus, subtle changes in the mitochondrial genotype can have profound effects on the nucleus, as well as carcinogenesis and cancer progression.

Circulating tumor DNA (ctDNA) and cfDNA can be isolated from the peripheral blood of ovarian cancer patients with reliable accuracy. Additionally, cfDNA variation has been identified in peripheral blood specimens [45,46,47,48]. Identifying genetic alteration originating from tumors in blood is called *liquid biopsy*, a technique that has been used in some cancers to drive individualized treatment, cancer surveillance, and early diagnosis, two of which (colorectal and lung) have recently gained FDA approval [49,50]. Given that ovarian cancer ctDNA is identifiable in patients with early-stage ovarian cancer, and that our model predicts ovarian cancer with good accuracy, the model described here has the potential to provide a good diagnostic tool for ovarian cancer at an early stage. Moreover, this method also has the potential to identify recurrent or persistent cancer in patients who have completed their adjuvant therapy, though the prediction model may have to be modified to account for those variations present in recurrent cancer. For the prediction model of ovarian cancer to be clinically ready to detect early-stage ovarian cancer, it would have to (1) be evaluated prospectively in an independent set and (2) include a varied number of ovarian cancers in different stages of the disease to capture all potential genetic variation.

One of the strengths of this study is that the mtDNA variation was detected with WES from samples of women with HGSC and from women with no pathology and no family history of ovarian cancer as controls. Additionally, variation analyses were performed following the recommended best practices of genome sequencing for all specimens, including the validation set. Testing of the prediction model was also performed in adequate cases and controls from the TCGA HGSC dataset, including only patients with the same tumor histological type as the initial analysis. Furthermore, we used different analytical platforms to validate the prediction model of cancer, achieving acceptable performances.

The number of samples in both cases and controls in our study may limit the generalizability of its results. The small sample size could both (a) limit the diversity of variants introduced in the study, excluding potential discriminant variables not found in our samples that may help improve the performance of the model, and (b) create a wide AUC 95% CI that may artificially augment model performance. Moreover, only one out of the 20 patients with ovarian cancer was black; the rest were white (1 unknown). This lack of diversity may make our pilot study not generalizable to other populations with more diverse backgrounds. While the ultimate goal of this study was to assess whether mtDNA variation could be used as a tool to identify ovarian cancer with the *liquid biopsy* technology, further analyses are needed to integrate both methodologies. To improve model performance, we need continued efforts to collect more specimens of varied stages, histologies, and increased racial diversity. Including more diverse samples will result in a more comprehensive mitochondrial variation mapping, which, with the help of modern analytics, will improve the accuracy and generalizability of the prediction model. Additionally, adding other markers (like CA125), clinical data, or genomic information to the model may improve its performance. In previous work aimed at improving the prediction of chemotherapy response in ovarian cancer [2,51,52], we observed that the integration of clinical, pathological, and genomic data improved the performance of prediction models. Ultimately, improved and robust prediction models for ovarian cancer detection will have to be validated prospectively in the context of multi-institutional clinical trials to capture all potential population variations and to ascertain their value and impact in early diagnosis of the disease, following the example of breast cancer [14].

## 4. Conclusions

In this retrospective, observational, pilot study, we identify a set of mtDNA variations that can discriminate HGSC with good performance. This effort could be the first step to a detection tool for ovarian cancer in serum, potentially even at early-stages. Additionally, this predictive model of HGSC has elements that are associated with expression of genes included in OXPHOS pathways with a potential cascade effect in cell biological functions.

## 5. Materials and Methods

We performed a single institution, retrospective, case–control pilot study using tumor specimens obtained at the time of cytoreductive surgery from 112 patients with HGSC (**cases**) and compared them to benign fallopian tube specimens from 14 patients collected at the time of surgery for benign indications (**controls**). DNA and RNA were isolated from all specimens. Whole-exome sequencing (WES) was performed in 20 cases and 14 controls, and RNA sequencing (RNAseq) was performed in 112 cases and 12 controls.

### 5.1. Specimen Acquisition

HGSC tissue samples and clinical outcome data were obtained from the Department of Obstetrics and Gynecology and Gynecologic Oncology Biobank (IRB, ID#200209010), which is a part of the Women’s Health Tissue Repository (WHTR, IRB, ID#201804817). All specimens archived in the Gynecologic Oncology Biobank (herein termed Biobank) were originally obtained from adult patients under written, informed consent in accordance with University of Iowa (UI) IRB guidelines. Tumor samples were collected, reviewed by a board-certified pathologist, flash-frozen, and then the diagnosis was confirmed in paraffin at the time of initial surgery. All experimental protocols were approved by the University of Iowa (UI) Biomedical IRB-01.

We collected fallopian tube samples from women undergoing gynecologic procedures. Fallopian tubes were obtained from patients with no family history of cancer besides squamous cell carcinoma of the skin, and who were undergoing salpingectomy for benign indications (mainly sterilization). Fallopian tubes were chosen as controls as this is the most likely origin of HGSC [53,54,55]. This approach has been used in the study of ovarian cancer and has been reported by our group and others [16]. DNA and RNA were extracted from epithelial tissue coming from the junction of the ampullary and fimbriated end of the fallopian tubes. Twenty normal fallopian tube specimens were obtained. Of those, 12 produced viable RNA for analysis. RNA from both the fallopian tube and HGSC specimens had already been extracted and purified in a previous study [56]. WES was performed on 14 fallopian tube specimens.

### 5.2. DNA Sequencing

Genomic DNAs (gDNAs) were purified from frozen tumor and fallopian tube tissues using the DNeasy Blood and Tissue Kit according to the manufacturer’s (QIAGEN, Hilden, Germany) recommendations. Yield and purity were assessed on a NanoDrop Model 2000 spectrophotometer and by using horizontal agarose gel electrophoresis. Whole-exome sequencing (WES) was performed externally by GeneWiz (Azenta, Chelmsford, MA, USA). Briefly, gDNA WES was performed on Illumina HiSeq 2000 (2 × 150 bp) with libraries prepared using Agilent SureSelect Human Exome Library Preparation V5 kit and sequenced to a 100× depth coverage. Raw reads were aligned to the human genome (hg38) using the Burrows–Wheeler Aligner [57]. The mean quality score (Phred) was 37.76, and the percentage of bases greater than or equal to 30 was 89.96. WES quality parameters for each normal control sample, including mtDNA coverage and read depth, are available in Appendix A. Depth of coverage was performed with *gatk* (v4.6.1.0), and quality control for high throughput sequencing with *FastQC* (v0.12.1).

### 5.3. RNA Sequencing

RNA was isolated from the tumor and control specimens in the Biobank. RNA extraction, processing, and sequencing have been described previously [51,58]. In brief, total cellular RNA was extracted from primary tumor tissue using the mirVana (Thermo Fisher, Waltham, MA, USA) RNA purification kit. The RNA yield and quality were assessed with a Trinean Dropsense 16 spectrophotometer and the Agilent Model 2100 bioanalyzer. The RNA quality was determined to be adequate if the sample had an RNA integrity number (RIN) of 7.0 or greater. Samples that were of adequate quality were then sequenced. In total, 500 ng of RNA was quantified by the Qubit measurement (Thermo Fisher). RNA was then converted to cDNA and ligated to sequencing adaptors with Illumina TriSeq stranded total RNA library preparation (Illumina, San Diego, CA, USA). cDNA samples were then sequenced with the Illumina HiSeq 4000 genome sequencing platform using 150 bp paired-end sequencing by synthesis (SBS) chemistry. All sequencing was performed at the Genome Facility at the University of Iowa Institute of Human Genetics (IIHG).

### 5.4. Single Nucleotide Variation (SNV) Analysis

DNA from WES was aligned to the mitochondrial DNA references sequence (Revised Cambridge Reference Sequence [rCRS] of the Human Mitochondrial DNA, or GenBank sequence NC_012920, https://www.mitomap.org/MITOMAP/HumanMitoSeq, accessed on 20 December 2024) using *BWA* (0.7.17-r1188). BAM files resulting from the alignment were used with *samtools* (v1.19.2) [59], the *Picard* toolkit (v2.27.1), and *gatk* (v4.6.1.0) [60] to create Variant Call Format (VCF) files for further analysis, as recommended by best practices of genome sequencing [61]. A table with present SNVs for all samples was constructed and used for the analyses.

To identify mitochondrial SNVs as being the most informative for HGSC prediction, we performed a multivariable LASSO regression analysis with all mtDNA variants using the *glmnet* R package (v4.1-8). The model was built with k-fold cross-validation (for internal validation), and the penalty parameter (λ) was set using bootstrapping to minimize overfitting with a small sample size [62]. We evaluated the performance of our model using the area under the curve (AUC) and its 95% confidence interval (CI). A value of 0.5 indicates a lack of model predictive performance, and 1.0 indicates perfect predictive performance. *LASSO* regression is aimed at prediction selection, not statistical inference. To estimate the inference and confidence intervals of the variables within the prediction model, we used the R package *selectiveInference* (v1.2.5), though the results may be subjected to overfitting.

### 5.5. Correlation of mtDNA Variables with Gene Expression and Pathway Analysis

STAR (v 2.7.11b) was used to align the RNAseq reads to the human reference genome (version hg38) [63]. We then created BAM files after alignment. The featureCount was used to measure gene expression [64]. The DESeq2 package (v1.38.3) was used to import, normalize, log2 transform, and prepare the gene counts for analysis [65]. ENSEMBL was used to annotate gene transcripts and identify sequence variants. Univariable analysis between both groups (cases and control) for all 37 mitochondrial genes with sequence variants was performed. Significant genes (adjusted *p*-value < 0.05) were introduced in a multivariable logistic regression analysis to determine which mitochondrial gene was independently associated with the cancer condition (*p*-value < 0.05). For differential gene expression analysis of the whole genome, we used a more restrictive FDR-adjusted statistical significance level (alpha) of 0.005.

To correlate SNV with gene expression, we used Spearman’s rank correlation test, as both variables are not completely independent one from each other. We assessed the statistical significance of the correlation by computing the *p*-value, using the false discovery rate (FDR) to correct for multiple analyses [66]. Using significant genes correlated with mtDNA SNVs, we performed an enrichment pathway analysis using the clusterProfiler R package (v4.3.3) [67], which interrogates the KEGG database (https://www.genome.jp/kegg/pathway.html, accessed on 17 December 2024). FDR-corrected *p*-values < 0.05 were considered statistically significant.

### 5.6. Prediction Model Testing

#### 5.6.1. Model Testing with the TCGA Database

Model testing was performed using the HGSC TCGA dataset. Briefly, after permission was granted to access controlled data by the Genomic Data Commons (GDC) Data Portal (dbGaP# 29868), TCGA HGSC WES data were downloaded for 448 HGSC samples (cases) and 190 control tissues collected from the same individuals. BWA, samtools, *Picard* toolkit (v2.27.1), and *gatk* (v4.6.1.0) were used to identify SNV and create VCF files for the analysis and testing of the prediction models of HGSC. Informative mtDNA SNVs from the UI prediction analysis were identified in the TCGA WES database and used to test the prediction model of HGSC. Initially, we created a prediction model of HGSC with the most informative SNVs using the *glmnet* R package (v4.1-8). Then, we selected the same informative SNVs in the TCGA database and applied the *pROC* (v1.18.5) R package to test the performance of the prediction model of HGSC in an independent dataset. 

Additionally, RNAseq (N = 423) files in BAM format were downloaded from TCGA for women with HGSC. We extracted gene expression data from these BAM files using STAR and *featureCounts* (v2.0.6). Significantly correlated genes within UI analyses were identified in the TCGA RNAseq dataset and then searched for associations with *MT-CYB* SNVs and expression alterations. We used the UI dataset gene expression in normal tubes (as controls) in combination with TCGA expression data for correlation validation. Data were normalized and log2 transformed before analysis. We used the same methods as described above for the UI data to correlate SNV with gene expression in TCGA data: Spearman’s rank correlation test, using FDR to correct for multiple analyses [66].

#### 5.6.2. Testing of Prediction Model of HGSC in Independent Analytical Platform

We used the MATLAB (v2024b) machine learning app (classification learner method) to test the prediction model of HGSC in a different analytical platform that leverages over 30 classifier methods. First, we trained and validated UI best prediction model of HGSC, including only the most informative SNVs. Then, we uploaded the same informative SNVs from TCGA to MATLAB and used these data to test the UI-trained and validated model. Testing was performed to account for the weights of the outcomes as well as for unbalanced data (fewer controls than cases).

## Figures and Tables

**Figure 1 ijms-26-01347-f001:**
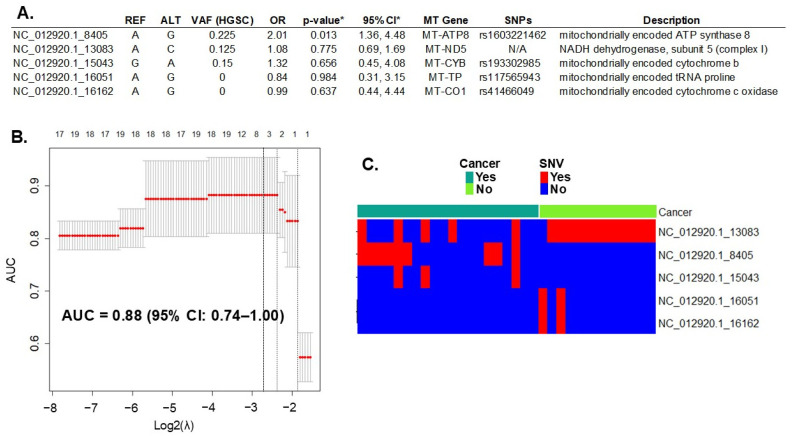
**Prediction of HGSC with mtDNA variation**. (**A**) Five mtDNA variants (out of 393 total variants) predicted ovarian cancer with a performance of 0.88 (95% CI of 0.74–1.00), measured by the area under the curve (AUC). REF: reference allele; ALT: alternative allele found; VAF: variant allele frequency of the allele found in cancer samples; OR: odds ratio between alleles found in cancer versus controls; MT Gene = name of the mitochondrial gene; SNP: known single nucleotide polymorphism number. * LASSO regression goal is to select variables for prediction, not for statistical inference. To estimate inference and confidence interval of the variant we used the R package *selectiveInference* (v. 1.2.5), though the results may be subjected to overfitting. (**B**) Graphic of the LASSO regression analysis with 95% CI. Top axis: number of variants (or single nucleotide variation—SNV) selected by the model; left axis: resulting AUC; bottom axis: log base 2 of lambda tunning parameter chosen with bootstrapping and cross-validation to optimize the model. The dotted lines represent the potential values: the one on the right is lambda.1se, in the middle is lambda.min, and on the left is the selected λ. (**C**) mtDNA variant count included in the model per each sample. Red: presence of SNV; Blue: absence of SNV. N/A, not applicable.

**Figure 2 ijms-26-01347-f002:**
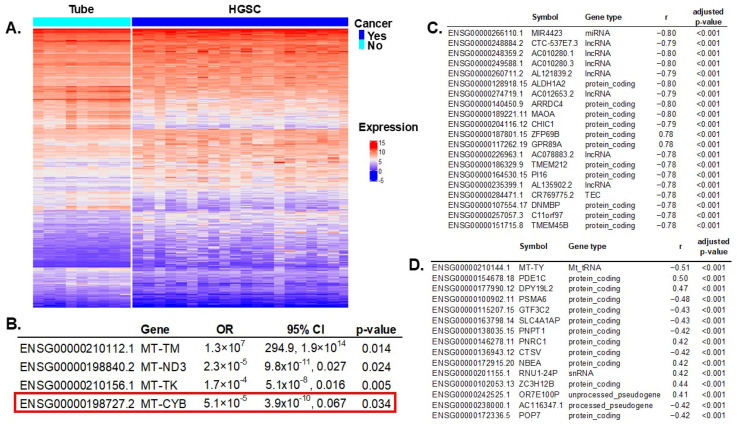
**Differential gene expression between HGSC vs. normal fallopian tube specimens and correlation with mtDNA variants.** (**A**) Differential gene expression between HGSC and tubal samples: 3382 transcripts out of 61,851 were significant, with a false discovery rate (FDR) adjusted *p*-value < 0.005 to account for multiple comparisons. (**B**) All 13 significant mitochondrial genes in the univariable differential expression analysis were introduced in a multivariable regression model; four were independently significant. Notably, *MT-CYB* (highlighted by the red box) also had SNVs that were characteristic of HGSC and were included in the prediction model of cancer. (Figure 1). (**C**) Correlation of significant gene expression (HGSC vs. control) and *MT-CYB* mtDNA genotype: 1728 significant genes (out of 3382) were significantly correlated (FDR-adjusted *p*-value < 0.005, to correct for multiple comparisons). Showing only the top 20. (**D**) Correlation of significant gene expression (HGSC vs. control) and *MT-CYB* mtDNA gene expression: 364 significant genes (out of 3382) were significantly correlated (FDR-adjusted *p*-value < 0.005). Showing only the top 15.

**Figure 3 ijms-26-01347-f003:**
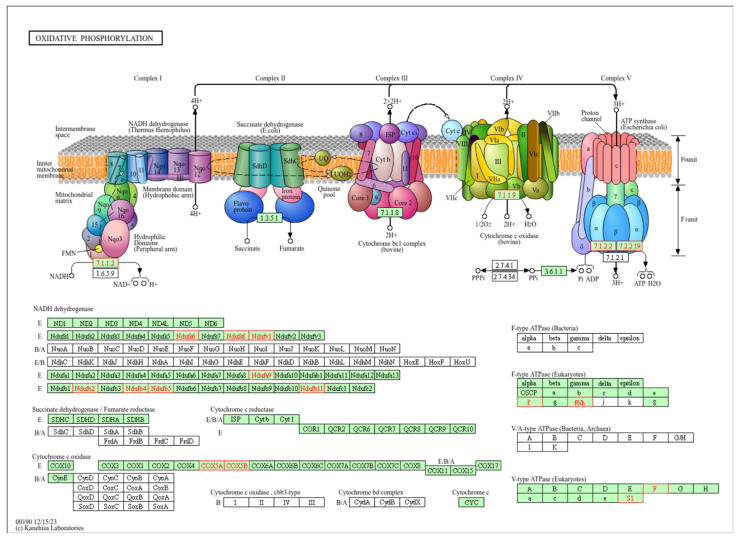
**Representation of the oxidative phosphorylation (OXPHOS) pathway.** Top-ranked pathway in the enrichment analysis (FDR adjusted *p*-value < 0.001). In the upper panel of this KEGG pathway (with permission), the components of the OXPHOS pathway, complexes I–IV, are represented in the mitochondrial membrane. In the lower panel, the components are detailed. Highlighted in red are the components that were significantly correlated with both *MT-CYB* genetic variation and expression in the pathway enrichment analysis. **Complex I (NADH dehydrogenase):** receives electrons from NADH and pumps protons across the mitochondrial membrane. **Complex II (succinate dehydrogenase):** receives electrons from succinate and does not directly contribute to proton pumping. **Complex III (ubiquinol cytochrome c reductase, bc1 complex):** accepts electrons from ubiquinone and pumps protons across the membrane. **Complex IV (cytochrome c oxidase):** receives electrons from cytochrome c and transfers them to oxygen to produce water, also pumping protons. **ATP synthase (Complex V):** uses the proton gradient established by the electron transport chain (ETC) to synthesize ATP from ADP and Pi.

**Figure 4 ijms-26-01347-f004:**
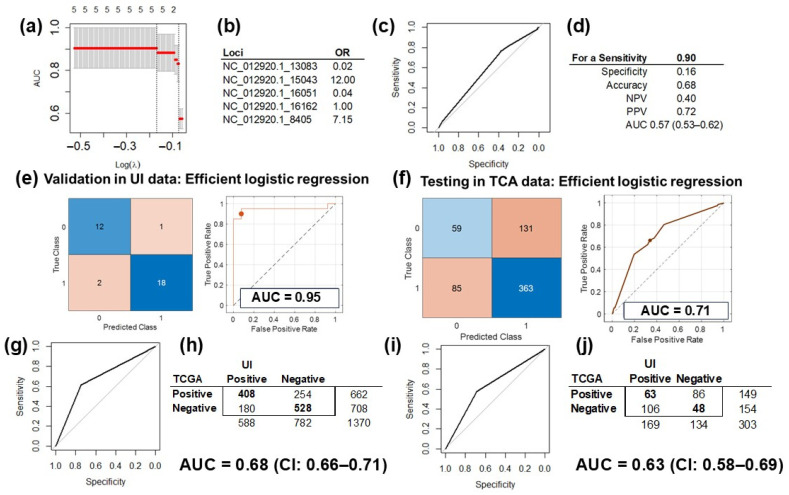
**Prediction model training, validation, and testing.** (**a**) Validation of the LASSO prediction model of HGSC using only the five informative mtDNA variants in the UI data (AUC = 91%, CI: 82–100%). (**b**) The odds ratio (OR) of the variants included in the LASSO model: >1 increases the risk for HGSC, and <1 is protective. (**c**) The ROC curve of the prediction model, tested on TCGA data with the pROC package. (**d**) Testing of the model was optimized for a sensitivity of 0.9 and performance measured in AUC at 0.57 (CI: 0.53–0.62). (**e**) Validation of the prediction model of HGSC using only the five informative mtDNA variants in UI data with *MATLAB*. Showing the confusion matrix (**left**) of the classified UI data, and the ROC curve for the validated model (**right**), with an AUC of 0.95 with the efficient logistic regression method. (**f**) Testing of the HGSC prediction model on TCGA data: confusion matrix (**left**) of the classified TCGA data, and the ROC curve for the tested model (**right**), with an AUC of 0.71 with the efficient logistic regression method. (**g**) Assessment of the correlation between the *MT-CYB* variant and differential gene expression of HGSC and controls (tubes) revealed a fair agreement between the results obtained for the UI and TCGA data (AUC = 0.68, CI: 0.66–0.71). (**h**) 2 × 2 matrix for the correlation assessed in Figure 4g: there were 1370 informative genes common for the UI and TCGA datasets. (**i**) Assessment of correlation between the *MT-CYB* expression and differential gene expression of HGSC and controls (tubes) revealed fair agreement between the results obtained for the UI and TCGA data (AUC = 0.63 CI: 0.58–0.69). (**j**) The 2 × 2 matrix for the correlation assessed in Figure 4i: there were 303 informative genes common for the UI and TCGA datasets.

**Table 1 ijms-26-01347-t001:** **Pathway enrichment analysis for genes, the expression of which was significantly correlated with MT-CYB gene variants and/or its expression.** Differentially expressed genes between HGSC and controls (N = 3382) were correlated with variations and gene expression of mitochondrial gene *MT-CYB*. All significantly (FDR-adjusted *p*-value < 0.005) correlated genes (1890) were introduced in a pathway enrichment analysis, resulting in significant pathways involved in cellular energy, cancer, and neurodegenerative disorders (FDR-adjusted *p*-value < 0.01).

ID	Category	Description	Fold	Adjusted *p*-Value
hsa00190	Energy metabolism	Oxidative phosphorylation	2.84	<0.001
hsa05415	Cardiovascular disease	Diabetic cardiomyopathy	2.29	<0.001
hsa04714	Environmental adaptation	Thermogenesis	2.17	<0.001
hsa05016	Neurodegenerative disease	Huntington disease	1.89	0.001
hsa05014	Neurodegenerative disease	Amyotrophic lateral sclerosis	1.80	0.001
hsa05012	Neurodegenerative disease	Parkinson disease	1.88	0.001
hsa04932	Endocrine and metabolic disease	Non-alcoholic fatty liver disease	2.12	0.003
hsa05020	Neurodegenerative disease	Prion disease	1.76	0.004
hsa05208	Cancer: overview	Chemical carcinogenesis - reactive oxygen species	1.81	0.006
hsa05010	Neurodegenerative disease	Alzheimer disease	1.55	0.009
hsa05022	Neurodegenerative disease	Pathways of neurodegeneration - multiple diseases	1.46	0.014
hsa00510	Glycan biosynthesis and metabolism	N-Glycan biosynthesis	2.59	0.017
hsa00600	Lipid metabolism	Sphingolipid metabolism	2.54	0.019
hsa04723	Nervous system	Retrograde endocannabinoid signaling	1.84	0.020

## Data Availability

Clinical data are not publicly available due to patient privacy. Datasets can be browsed in the Gene Expression Omnibus (GEO) database by their accession number: GSE156699. The validation part of this study was performed in silico, with de-identified publicly available data. All data from TCGA is available at their website: https://portal.gdc.cancer.gov/, accessed on 20 December 2024. Software utilized by this study is also publicly available at Bioconductor website: http://bioconductor.org/, accessed on 20 December 2024.

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
