# Peer review of "Identification of Ovarian High-Grade Serous Carcinoma with Mitochondrial Gene Variation"

_ijms, 2025, doi:10.3390/ijms26031347_

Round 1
Reviewer 1 Report
Comments and Suggestions for Authors
The manuscript "Identification of ovarian high-grade serous carcinoma with mitochondrial gene variation" explores a compelling new avenue for improving the early detection of ovarian cancer, a disease notorious for being diagnosed at advanced stages. By focusing on mtDNA variations, the authors investigate whether these genetic markers can distinguish HGSC – the most common and aggressive form of ovarian cancer – from normal tissues.
In this pilot study, the research team analyzed 20 HGSC samples and 14 normal controls using WES. They identified specific mtDNA single nucleotide variants SNVs that appear to correlate with cancer risk. Their predictive model, built from five key mtDNA variants, demonstrated strong performance with an AUC of 0.88. Among the most significant findings was variation in the MT-CYB gene, which not only increased the likelihood of HGSC by more than 30% but also showed reduced expression in cancer samples.
The study stands out for its innovative approach to addressing a major clinical challenge. By leveraging genetic data from mitochondrial DNA, the authors present a promising step toward developing a non-invasive diagnostic tool that could potentially be implemented as a liquid biopsy.
However, the study is not without its limitations. The relatively small sample size raises questions about the generalizability of the findings, and while the model performed well initially, its validation in external datasets yielded more modest results (with AUCs ranging from 0.63 to 0.71). Expanding the study to include larger, more diverse cohorts will be crucial for confirming the robustness of the approach and enhancing its clinical relevance.
Some suggestions:
- Include supplementary tables listing all mtDNA variants and their clinical associations.
- Emphasize clinical relevance in the abstract and highlight the potential for liquid biopsy integration.
- Expand citations to include recent advances in liquid biopsy and cfDNA research.
- Clearly outline how the study advances the field and propose next steps like machine learning integration or multi-center trials.
Author Response
Comment 1: Include supplementary tables listing all mtDNA variants and their clinical associations.
Response 1: Supplementary Table S1 was added. Doing an association of mitochondrial variation with clinical characteristics was beyond the scope of this study.
Comment 2: Emphasize clinical relevance in the abstract and highlight the potential for liquid biopsy integration.
Response 2: Added to the end of the Abstract.
Comment 3: Expand citations to include recent advances in liquid biopsy and cfDNA research.
Response 3: Both, advances and references, have been added to the Introduction.
Comment 4: Clearly outline how the study advances the field and propose next steps like machine learning integration or multi-center trials.
Response 4: At the end of the Discussion, we added our experience with data integration and effect on prediction modelling as well as application in clinical trials.
Reviewer 2 Report
Comments and Suggestions for Authors
Dear Editor,
I read with great interest the article entitled “Identification of ovarian high-grade serous carcinoma with mitochondrial gene variation” by Jesus Gonzalez Bosquet et al., submitted to the International Journal of Molecular Sciences.
As the title suggests, the study aims to explore high-grade serous carcinoma (HGSC) of the ovary through mitochondrial gene variations, given that no current screening method has proven effective in detecting ovarian cancer at earlier stages.
To summarize, the main objectives of the study were:
To assess whether mtDNA variations can distinguish HGSC from normal tissue.
To validate HGSC predictive models based on mtDNA variations across different platforms and datasets, including the TCGA HGSC dataset.
To investigate correlations between significant mtDNA variants and gene expression in HGSC.
To determine whether gene patterns associated with mtDNA variants are linked to clinical outcomes in HGSC.
For objective 1, the authors used whole-exome sequencing (WES) with germline-level coverage and depth in unpaired samples of HGSC tumor tissue (20 samples) and normal fallopian tube tissue (14 samples) from different individuals. The identified SNVs were analyzed using a LASSO multivariate regression, which selected 5 mitochondrial variants deemed informative for HGSC prediction. A predictive model using these 5 variants achieved an AUC of 0.88 (95% CI: 0.74–1.00). However, there is no mention of the VAF of these variants or whether they are present in population databases and their respective frequencies. Only the odds ratios (OR) were reported, with no 95% confidence intervals, making it difficult to assess the statistical significance of these findings. Without this information and given the comparison of unpaired samples, it is impossible to determine whether the variants are germline or somatic. The fact that they were sequenced in tumors alone does not clarify their origin. Moreover, the frequency differences observed between cases and controls, resulting in the AUC of 0.88, may merely reflect differences in the genetic constitution of individuals rather than being a tumor-specific consequence.
If these variants are germline and collectively confer an increased risk of HGSC, the small sample size tested lacks sufficient statistical power. This limitation is further compounded by the absence of 95% confidence intervals and p-values for the reported ORs, leaving only AUC values to evaluate the model. This lack of clarity undermines any conclusions related to objective 1. The inability to determine whether the variants observed in tissues are germline or somatic significantly compromises the study's goal of distinguishing HGSC from normal tissue. Furthermore, this uncertainty also affects the proposed application of the findings, such as liquid biopsies. Without confirmation that these variants are somatic, it is difficult to justify the analysis of ctDNA for tumor detection. This lack of clarity is a critical methodological issue that compromises the proposed model and its intended applications.
Regarding objective 2, the results presented starting at line 121 follow the analysis of gene expression in 20 HGSC samples and 14 normal fallopian tube samples. However, the text is disorganized and difficult to follow, making it challenging to interpret the findings. There appear to be missing words in this section, and the presence of large blank spaces further disrupts the flow of information, making it unclear what the authors are trying to communicate. These issues remain unresolved even after consulting the methods section, which lacks sufficient detail about the approach used.
In the methods, the authors state that the TCGA data used for validation consisted of 448 HGSC samples (cases) and 190 control tissue samples collected from the same individuals. The WES data were processed with the same tools as before, and the same 5 mitochondrial SNVs were used to validate the predictive model with TCGA data. However, the results present two sets of AUCs: one with high values (0.91 and 0.95) and another with significantly lower values (0.57 and 0.71). The methods section does not adequately explain how the TCGA data were partitioned to produce such discrepant results. The lack of clarity in the methods, coupled with the apparent omissions and large blank spaces in the text, impairs the understanding of the results and compromises the transparency and reproducibility of the study.
The overall presentation of the article—including the introduction, results, methods, discussion, and conclusion—is confusing, with figures adding to the disorganization. This significantly hinders the reader's ability to fully grasp the study's findings. I strongly recommend that the authors review and restructure their manuscript to ensure it is clearer and better organized. This lack of methodological clarity undermines both the transparency and reproducibility of the study.
Given the methodological issues with objective 1 and the disorganization in the presentation of objective 2, my recommendation is to reject the study in its current form.
Below, I provide some additional suggestions to help the authors improve their manuscript for future submissions:
In Figure 1A, the authors could include 95% confidence intervals (CI) and p-values for each odds ratio (OR) of the 5 mtDNA SNVs to clarify the statistical significance of the findings.
Discuss the validity of comparing the gene expression of normal fallopian tubes and HGSC samples to demonstrate the robustness of this approach. Include this discussion in the results or discussion sections, referencing item 2.2, and clarify whether similar expression patterns would be expected in normal tissue types.
Regarding higher gene expression in tumor samples, it would be valuable to address whether increased mitochondrial activity is already expected in cancerous conditions.
Ensure that all acronyms are defined the first time they appear in the text (e.g., FDR).
Reevaluate whether Figure 3 is necessary for the article.
Provide more details about the exome used, including information about the manufacturer, sequencing equipment, and technical parameters of the sequencing performed at GeneWiz (Azenta, Chelmsford, MA).
Since exome sequencing in tissue is uncommon, provide a detailed description of quality parameters for each sample, including mean and minimum mtDNA coverage.
Include a more detailed description of the classifier developed for HGSC detection, as referenced in "We built a classifier to detect HGSC as detailed previously [16]." Specify parameters, algorithms, and evaluation strategies.
Revise and expand the descriptions in section 5.6 to ensure clarity and reproducibility. Provide consistent details about the tools and methodologies used, including complete tool names and versions (e.g., "BWA vX.X"), and clarify statistical validation methods.
Ensure references are complete, including the names of the journals where cited articles were published.
Author Response
Comment 1. For objective 1, the authors used whole-exome sequencing (WES) with germline-level coverage and depth in unpaired samples of HGSC tumor tissue (20 samples) and normal fallopian tube tissue (14 samples) from different individuals. The identified SNVs were analyzed using a LASSO multivariate regression, which selected 5 mitochondrial variants deemed informative for HGSC prediction. A predictive model using these 5 variants achieved an AUC of 0.88 (95% CI: 0.74–1.00). However, there is no mention of the VAF of these variants or whether they are present in population databases and their respective frequencies. Only the odds ratios (OR) were reported, with no 95% confidence intervals, making it difficult to assess the statistical significance of these findings. Without this information and given the comparison of unpaired samples, it is impossible to determine whether the variants are germline or somatic. The fact that they were sequenced in tumors alone does not clarify their origin. Moreover, the frequency differences observed between cases and controls, resulting in the AUC of 0.88, may merely reflect differences in the genetic constitution of individuals rather than being a tumor-specific consequence.
If these variants are germline and collectively confer an increased risk of HGSC, the small sample size tested lacks sufficient statistical power. This limitation is further compounded by the absence of 95% confidence intervals and p-values for the reported ORs, leaving only AUC values to evaluate the model. This lack of clarity undermines any conclusions related to objective 1. The inability to determine whether the variants observed in tissues are germline or somatic significantly compromises the study's goal of distinguishing HGSC from normal tissue. Furthermore, this uncertainty also affects the proposed application of the findings, such as liquid biopsies. Without confirmation that these variants are somatic, it is difficult to justify the analysis of ctDNA for tumor detection. This lack of clarity is a critical methodological issue that compromises the proposed model and its intended applications.
Response 1: We apologize if the manuscript was not clear. It seems that after submission the editorial team changed the Methods section to the end and put the Results section after the Introduction. That is not how we initially designed the manuscript so there are some inconsistencies in how things were described. We will review the whole manuscript to make it more legible. Also, there are several issues that we would like to address:
- As the reviewer noted, the best way to determine whether variants of nuclear DNA are germline or somatic is comparing sequenced paired samples from the same individual, one originated from tumor and another from normal tissue. However mitochondrial DNA (mtDNA) has especial considerations because is highly susceptible to acquiring mutations from reactive oxygen species (ROS), even in normal tissues. Therefore, multiple subpopulations of mtDNA could arise during the mitochondria lifespan, this is the phenomenon known as heteroplasmy (Jimenez-Morales, S.; Perez-Amado, C. J.; Langley, E.; Hidalgo-Miranda, A., Overview of mitochondrial germline variants and mutations in human disease: Focus on breast cancer (Review). Int J Oncol 2018, 53, (3), 923-936). Heteroplasmy describes the situation in which two or more mtDNA variants exist within the same cell. These variants can occur either in the germline or in the tumoral tissues and can vary from cell to cell. So up to 72% of the reported tumor-specific mtDNA mutations are also found in non-tumor cells (germline variants) of healthy subjects (Larman, T. C.; DePalma, S. R.; Hadjipanayis, A. G.; Cancer Genome Atlas Research, N.; Protopopov, A.; Zhang, J.; Gabriel, S. B.; Chin, L.; Seidman, C. E.; Kucherlapati, R.; Seidman, J. G., Spectrum of somatic mitochondrial mutations in five cancers. Proceedings of the National Academy of Sciences of the United States of America 2012, 109, (35), 14087-91).
Heteroplasmy is the reason why we decided to use all variants, without considering origin, to build our model of ovarian cancer prediction. The model itself will discriminate which variants determine the risk for cancer, even if they are derived from mutations in normal tissue. We added this rationale to the discussion and the references to the manuscript.
- As the reviewer appreciates, lasso regression models are built for prediction and measured by c-statistics, including the area under the (receiver operating characteristic) curve (AUC), not for inference. There is a way of determining confidence and inference, although it must be done after the model selection, it must assume that the lambda is fixed and could add overfitting to the model. It was added to the Results (Figure 1) and was discussed later.
- Variant allele frequency (VAF) for the variants included in the model were added in Figure 1A.
Comment 2: Regarding objective 2, the results presented starting at line 121 follow the analysis of gene expression in 20 HGSC samples and 14 normal fallopian tube samples. However, the text is disorganized and difficult to follow, making it challenging to interpret the findings. There appear to be missing words in this section, and the presence of large blank spaces further disrupts the flow of information, making it unclear what the authors are trying to communicate. These issues remain unresolved even after consulting the methods section, which lacks sufficient detail about the approach used.
Response 2: Again, we apologize for these inconsistencies derived from reshuffle of the text by the editorial team: placing Methods section and the end, ahead of Results. We reviewed the text to make it more understandable.
Comment 3: In the methods, the authors state that the TCGA data used for validation consisted of 448 HGSC samples (cases) and 190 control tissue samples collected from the same individuals. The WES data were processed with the same tools as before, and the same 5 mitochondrial SNVs were used to validate the predictive model with TCGA data. However, the results present two sets of AUCs: one with high values (0.91 and 0.95) and another with significantly lower values (0.57 and 0.71). The methods section does not adequately explain how the TCGA data were partitioned to produce such discrepant results. The lack of clarity in the methods, coupled with the apparent omissions and large blank spaces in the text, impairs the understanding of the results and compromises the transparency and reproducibility of the study.
Response 3: We clarified the Methods and Results sections to make it more understandable. Then, we discussed those results to explain differences in model performances. We addressed different AUC values and how they compare based on their 95% CI.
Comment 4: Below, I provide some additional suggestions to help the authors improve their manuscript for future submissions:
Comment 4.1. In Figure 1A, the authors could include 95% confidence intervals (CI) and p-values for each odds ratio (OR) of the 5 mtDNA SNVs to clarify the statistical significance of the findings.
Response 4.1. Done.
Comment 4.2. Discuss the validity of comparing the gene expression of normal fallopian tubes and HGSC samples to demonstrate the robustness of this approach. Include this discussion in the results or discussion sections, referencing item 2.2, and clarify whether similar expression patterns would be expected in normal tissue types.
Response 4.2. Fallopian tubes are the most likely origin of ovarian cancers, specifically the HGSC type (see references 53-55). That is the reason why they were chosen as controls for analysis of variation and gene expression. We added a comment in the Methods section and Discussion, with additional references.
Comment 4.3. Regarding higher gene expression in tumor samples, it would be valuable to address whether increased mitochondrial activity is already expected in cancerous conditions.
Response 4.3. We added further discussion on mitochondrial gene expression in cancers.
Comment 4.4. Ensure that all acronyms are defined the first time they appear in the text (e.g., FDR).
Response 4.4. Done.
Comment 4.5. Reevaluate whether Figure 3 is necessary for the article.
Response 4.5. We believe so, being the most important pathway affected by mtDNA variation and changes in gene expression in HGSC.
Comment 4.6. Provide more details about the exome used, including information about the manufacturer, sequencing equipment, and technical parameters of the sequencing performed at GeneWiz (Azenta, Chelmsford, MA).
Response 4.6. Added to the Methods section.
Comment 4.7. Since exome sequencing in tissue is uncommon, provide a detailed description of quality parameters for each sample, including mean and minimum mtDNA coverage.
Response 4.7. WES quality parameters for each control (tubal) sample, including mtDNA coverage and read depth are available in Supplementary Table S2.
Comment 4.8. Include a more detailed description of the classifier developed for HGSC detection, as referenced in "We built a classifier to detect HGSC as detailed previously [16]." Specify parameters, algorithms, and evaluation strategies.
Response 4.8. Details added to Methods section
Comment 4.9. Revise and expand the descriptions in section 5.6 to ensure clarity and reproducibility. Provide consistent details about the tools and methodologies used, including complete tool names and versions (e.g., "BWA vX.X"), and clarify statistical validation methods.
Response 4.9. We specified the methods used for validation of all analyses: prediction model of ovarian cancer with TCGA data with R environment and MATLAB analytic platform, and correlation of SNV and gene expression in TCGA dataset.
Comment 4.10. Ensure references are complete, including the names of the journals where cited articles were published.
Response 4.10. References have been reformatted and revised following the IJMS template of EndNote.
Reviewer 3 Report
Comments and Suggestions for Authors
This paper proposes a prediction model based on mitochondrial DNA variation for identifying well-differentiated ovarian serous carcinoma (HGSC) and verifies it. The study is innovative, especially in the field of combining mitochondrial gene variation with cancer detection. However, there are still some contents that need to be further improved to improve the scientificity and completeness of the study.
Here are some suggestions for improving this article:
l Add analysis of model performance in different stages of ovarian cancer: It is recommended to add an analysis of the model's predictive performance in different stages of ovarian cancer to better assess its potential for actual clinical application.
l Discuss how to combine the model with existing diagnostic methods: It is suggested to discuss in the discussion section how to combine the prediction model based on mitochondrial DNA variation with existing diagnostic methods (such as CA-125 marker detection) to improve predictive accuracy.
l Explain the biological significance of the 20 genes related to MT-CYB in Figure 2: Figure 2 lists 20 genes related to MT-CYB but does not explain their biological functions. It is recommended to briefly describe the functions of these key genes in the discussion.
l Clearly discuss the impact of sample size and racial diversity on the predictive model: The small sample size, especially with only 14 normal controls, may limit the generalizability of the results. It is recommended to clearly point out in the discussion the impact of the small sample size and lack of racial diversity on the performance of the predictive model, and emphasize the importance of expanding sample size and increasing racial diversity in future studies.
l Add more literature support for the association between MT-CYB variation and cancer: Although the text mentions that MT-CYB variation significantly increases the risk of HGSC, it does not fully discuss its biological mechanism. It is suggested to add more relevant literature support in the discussion, emphasizing the association between this gene and cancer, to enhance the persuasiveness of the study's conclusions.
Author Response
Comment 1. Add analysis of model performance in different stages of ovarian cancer: It is recommended to add an analysis of the model's predictive performance in different stages of ovarian cancer to better assess its potential for actual clinical application.
Response 1. Ideally, we would do as the reviewer says. However, this was a pilot study to assess whether the concept could be developed further. We used the most aggressive type of ovarian cancer, high grade serous and all of them in advanced stage: 12 stages III and 8 stages IV. We will add this information to the Methods section.
Comment 2. Discuss how to combine the model with existing diagnostic methods: It is suggested to discuss in the discussion section how to combine the prediction model based on mitochondrial DNA variation with existing diagnostic methods (such as CA-125 marker detection) to improve predictive accuracy.
Response 2. At the end of the Discussion we added a few sentences about data integration for prediction modelling building.
Comment 3. Explain the biological significance of the 20 genes related to MT-CYB in Figure 2: Figure 2 lists 20 genes related to MT-CYB but does not explain their biological functions. It is recommended to briefly describe the functions of these key genes in the discussion.
Response 3. We have added further discussion about the functional repercussions of these top genes in Figure 2. We placed with the discussion of the OXPHOS pathways and other differential gene expression in HGSC correlated with mtDNA variation.
Comment 4. Clearly discuss the impact of sample size and racial diversity on the predictive model: The small sample size, especially with only 14 normal controls, may limit the generalizability of the results. It is recommended to clearly point out in the discussion the impact of the small sample size and lack of racial diversity on the performance of the predictive model, and emphasize the importance of expanding sample size and increasing racial diversity in future studies.
Response 4. Sample size and racial diversity were added to the Discussion.
Comment 5. Add more literature support for the association between MT-CYB variation and cancer: Although the text mentions that MT-CYB variation significantly increases the risk of HGSC, it does not fully discuss its biological mechanism. It is suggested to add more relevant literature support in the discussion, emphasizing the association between this gene and cancer, to enhance the persuasiveness of the study's conclusions.
Response 5. Both the biological mechanism as well references were added to the Discussion
Reviewer 4 Report
Comments and Suggestions for Authors
Abstract
· First page line11: Replace “that” with “than”.
· First page Line 14: Replace “DNA mitochondrial” with “mitochondrial DNA”.
· First page Line 22: What is the sentence “fair performance (between 0.63-0.71)” refers to or what is performance it describes here?
· First page Line 22: Replace “identify” with “identified”.
Introduction
· First page line 38: Replace “from” with “in”.
· Why the authors discussed the concept of cfDNA and liquid biopsy although the samples understudy were obtained from tumor tissues?
Discussion
· The authors stated that “While the primary objective was to probe whether mtDNA variation could be used as a tool to identify ovarian cancer with the liquid biopsy technology, further analyses are needed to integrate both methodologies”.
My question: how this paper is related to liquid biopsy concept by any means??
· In page 8 line 249: Please revise this phrase “To further these results” as it is meaningless.
Materials and Methods
· The authors should provide more details related to how they extracted mtDNA from the tumor tissue and control and mention the kits used in this procedure.
· In page 8 Line 287: Delete number 5 before the subtitle “5DNA sequencing”.
Author Response
Abstract
Comment 1. First page line11: Replace “that” with “than”.
Response 1. Done
Comment 2. First page Line 14: Replace “DNA mitochondrial” with “mitochondrial DNA”.
Response 2. Done
Comment 3. First page Line 22: What is the sentence “fair performance (between 0.63-0.71)” refers to or what is performance it describes here?
Response 3. Edited to clarify performance in terms of AUC.
Comment 4. First page Line 22: Replace “identify” with “identified”.
Response 4. Done
Introduction
Comment 1. First page line 38: Replace “from” with “in”.
Response 1. Done
Comment 2. Why the authors discussed the concept of cfDNA and liquid biopsy although the samples understudy were obtained from tumor tissues?
Response 2. Because there is the potential to apply the results of this study in a future test for early detection of ovarian cancer. We added the rationale to the Introduction.
Discussion
Comment 1. The authors stated that “While the primary objective was to probe whether mtDNA variation could be used as a tool to identify ovarian cancer with the liquid biopsy technology, further analyses are needed to integrate both methodologies”.
My question: how this paper is related to liquid biopsy concept by any means??
Response 1. Because if we can build a model that uses genetic variation with/out other significant biomarkers and that discriminates correctly ovarian cancer from normal individuals, we may be able to use it in the diagnosis of ovarian cancer with the liquid biopsy methodology.
Comment 2. In page 8 line 249: Please revise this phrase “To further these results” as it is meaningless.
Response 2. Changed to ‘To improve model performance’
Materials and Methods
Comment 1. The authors should provide more details related to how they extracted mtDNA from the tumor tissue and control and mention the kits used in this procedure.
Response 1. This was also requested by Reviewer #2. We added this to Methods.
Comment 2. In page 8 Line 287: Delete number 5 before the subtitle “5DNA sequencing”.
Response 2. Done
Round 2
Reviewer 1 Report
Comments and Suggestions for Authors
thanks for having considered my comments
Author Response
Thank you
Reviewer 2 Report
Comments and Suggestions for Authors
The authors did an excellent job addressing my initial questions and adjusting the presentation of the article. I have no further comments and recommend acceptance for publication.
Author Response
Thank you